# Changes in the Mental Health Indicators and Training Opportunities for Estonian Elite Athletes Compared to the COVID-19 Isolation Period

**DOI:** 10.3390/sports10050076

**Published:** 2022-05-11

**Authors:** Anna-Liisa Tamm, Ülle Parm, Anu Aluoja, Tuuli Tomingas

**Affiliations:** 1Physiotherapy and Environmental Health Department, Tartu Health Care College, 50411 Tartu, Estonia; ylleparm@nooruse.ee (Ü.P.); tuulitomingas@nooruse.ee (T.T.); 2Department of Psychiatry, Faculty of Medicine, University of Tartu, 50417 Tartu, Estonia; anu.aluoja@ut.ee

**Keywords:** athletes, coronavirus, anxiety, depression

## Abstract

*Background*: In spring 2020, two-thirds of Estonian elite athletes had symptoms of emotional distress. The aim of this study was to evaluate the mental health indicators and training opportunities for elite Estonian athletes a year after the complete COVID-19 isolation period compared to June 2020. *Methods*: In both cross-sectional studies, athletes completed self-reported questionnaires, including the Emotional State Questionnaire. Descriptive statistics, *t*-tests, and Chi2 tests were applied to compare the study groups (*p* < 0.05). *Results*: A total of 172 out of approximately 600 elite Estonian athletes participated in the survey (102 in 2020 and 70 in 2021). More than a year after the COVID-19 lockdown period, the mental health problems of elite athletes (particularly the symptoms of depression and fatigue) are even greater concern than in June 2020, despite the recovery in training conditions and competition. Of all of the subjects, 80% had high levels of distress in 2021 compared to 36% in 2020. According to the athletes, the availability of health care services was good (78.6%), but there was a lack of close cooperation with the coach. However, the athletes considered their coaches to be their main supporters, along with their family members and partners. Only 4.3% of the respondents considered a sports psychologist to be their main supporter (*n* = 6). *Conclusions*: More than a year after the COVID-19 lockdown period, the mental health indicators of Estonian elite athletes were worrisome. Most of subjects had high levels of distress even though their training conditions had returned to normal (i.e., to as they had been before COVID-19).

## 1. Introduction

In the spring of 2020, due to the rapid spread of SARS-CoV-2 (COVID-19), many countries announced a national lockdown (including closing sporting facilities), and national and international title competitions were postponed or cancelled. Isolation had important implications for the mental health of the general population [1,2,3], especially athletes [4,5]. The restriction period caused significant changes in athletes’ training routines [6], and although efforts were made to maintain physical activity, several studies showed a reduction in athletes’ training frequency, duration [7] and intensity, mostly because athletes and their coaches did not have enough time to prepare training programs for in-home training [8]. For example, 50% of swimmers and rowers on the Spanish Olympic team said that despite being able to take advantage of home training conditions, they lost quite a lot or a lot of their physical fitness during the isolation period [9]. Decreased physical activity had a negative effect on athletic performance, which in turn increased stress levels and created negative mood states [10]. Symptoms of stress and depression also increased due to insufficient personal contact with their coaches and psychologists [11,12].

In spring 2020, the findings on Estonian elite athletes’ mental health were also worrisome, as two-thirds of athletes had symptoms of emotional distress [13]. However, it must be acknowledged that despite the cancellation of competitions and changes in training conditions, the athletes remained generally positive, and it was hoped that the pandemic would have a modest impact on Estonian sports [14]. A similar result was obtained by Moscoso-Sanchez et al. [9], who studied the swimmers and rowers who represented Spain at the Olympic Games.

In the spring of 2021 (a year after the COVID-19 lockdown period), gyms were opened in Estonia, no state restrictions were imposed, and vaccination against COVID-19 was available to all adult Estonians. Many competitions took place around the world. The sports crowd was anticipating the postponed 32nd Summer Olympic Games although, according to the International Olympic Committee’s (IOC) order, the games were to be held mostly without the participation of spectators [15]. Therefore, it is important to know how and whether the quick and unpredictable changes in the COVID-19 situation influenced athletes’ opportunities and affected their long-term mental health. The aim of the study was to evaluate the mental health indicators and training opportunities for elite Estonian athletes a year after the complete COVID-19 isolation period compared to June 2020.

## 2. Materials and Methods

The Committee of Ethics of the University of Tartu approved the study (protocol no. 340/T-4, 19 April 2021). The selection of subjects was based on Swann et al. [16] with regard to the definition of an elite athlete: an athlete who competes at the national or international level, including in the home league. The questionnaire (via electronic system connect.ee) was sent to all Estonian elite athletes (approximately 600), who have represented Estonia in last two years in national and international title competitions in the adult age group by the contact person of the Estonian Olympic Committee at the beginning of June 2021. As minors (from the age of 16) can also participate in the adult age group in certain sports (such as gymnastics), they were also included in the study. The questionnaire was anonymous and there was no need or opportunity to identify respondents. There was therefore no need to sign any informed consent. At the beginning of the questionnaire, the purpose of the study and the issues related to confidentiality were explained to the subjects, and the subject could contact the responsible investigator if they had any questions. The athletes were given two weeks to complete the questionnaire. The questionnaire took approximately 20 min, and participants could withdraw at any time.

The design of the study was analogous to the study conducted in spring 2020 [13], but the study group was not the same. The self-assessment questionnaire was compiled by the authors of the article and was piloted before 2020, in addition to a 2021 survey of five elite Estonian athletes. The questionnaire collected data on the competitions currently being prepared, training conditions and opportunities (including comparisons to pre-COVID-19 and the lockdown period), attitudes towards and visits to health professionals, and mental health indicators such as depression, anxiety, fatigue, and insomnia (the Emotional State Questionnaire, EST-Q2) [17,18]. In the EST-Q2, participants reported how much various problems had troubled them during the past four weeks, using the scale 0 = not at all, 1 = seldom, 2 = sometimes, 3 = often, 4 = all the time. The instrument helps to identify symptoms related to mental health, but it is not meant to diagnose mental disorders. The results were compared with the results of a survey conducted in June 2020, after the isolation period, but relating to the COVID-19 isolation period [13,14]. As the authors of the study, we understand that the collected data reflects only the subjective experiences and views of the athletes regarding training.

The software program Sigma Plot for Windows version 11.0 (GmbH Formation, Germany) was used. The whole study group was divided into subgroups according to their EST-Q2 scores: (1) low distress—none of the subscale scores (depression, anxiety, fatigue, and insomnia) surpassed the referent value, (2) moderate distress—some scores were on the cutoff value and/or one surpassed it, (3) high distress—most subscale values were over the cutoff value [13]. Results are presented as means with standard deviation (SD), percentages, or odds ratios (OR) with 95% confidence intervals (CI). Continuous variables were compared with the t-test or Mann-Whitney test, and categorical with the Chi-squared or Fisher’s exact test, as appropriate. In comparing athletes’ opinions about the most disturbing consequences of COVID-19, the low distress group was not included, as there were only four representatives. A *p* value of < 0.05 was considered significant.

## 3. Results

### 3.1. Baseline Characteristics

In total, 172 elite athletes completed the questionnaire, 102 in 2020 and 70 athletes in 2021. The results of the study carried out in 2020 have been presented previously [3,4], but we have presented some of these results here as well to determine if there has been a change between the two study periods. In 2021, the respondents represented ~12% of all invited participants (Estonian elite athletes; ♀ = 46), with an average age of 22.36 ± 7.28 years (including 23 minors, i.e., <18 years). Of the 2021 participants, 77.1% participated in individual sports, and 52.9% had won a place in the top six in international junior or adult title competitions. Most of the respondents (92.9%) trained in Estonia, and 17.1% prepared for the Olympic Games, 30% for the World Championships, and 42.9% for the European Championships. By June 2021, 77% of subjects had been diagnosed with COVID-19, and 67.14% had received a primary vaccine against COVID-19. In total, 48.6% of athletes had been both diagnosed with COVID-19 and vaccinated.

### 3.2. Symptoms Related to Mental Health

The gender and age of different EST-Q2 groups in both studies are presented in Table 1. In 2021, 80% of the subjects had high distress according to the EST-Q2 questionnaire, which is more than twice as much as in the 2020 survey. There were no differences in participants’ age in the different research groups (♂ vs. ♀ *p* = 0.058) and years. In the 2021 survey, according to EST-Q2, 56 athletes belonged to the high distress group, while only four athletes belonged to the low distress group and 10 athletes to the moderate distress group (Table 1). In 2021, the chance of belonging to the low distress group was eight times lower (OR = 7.89; 95% CI 2.65–23.5), the chance of belonging to the moderate distress group was three-times lower, and the chance of belonging to the high distress group was seven-times higher (OR = 7.03; 95% CI 3.45–14.31) compared with 2020. Among athletes, there were significantly more women who belonged to the high distress group than men (*p* < 0.001). Compared to June 2020, at the beginning of summer 2021, the majority of athletes belonged to the high distress group.

The prevalence of symptoms related to mental health in elite Estonian athletes in 2020 and in 2021 are presented in Figure 1. The results show that at the beginning of the summer of 2021, the most common symptoms in female and male athletes were depression and fatigue, and fatigue and insomnia, respectively. Moreover, the incidence of depression and anxiety symptoms was higher in women than in men. Over the year, the symptoms of depression, anxiety, and fatigue significantly increased in both female and male athletes. In 2021, 32 (45.7%) subjects had above-cutoff scores in all four subcategories.

### 3.3. Training Conditions

In 2021, the subjects indicated their main training places before the COVID-19 pandemic, both in the spring of 2020 (lockdown in Estonia), and in the spring of 2021, when the training facilities were already open for athletes (Figure 2). Training opportunities were normalized compared to June 2020 and resembled the pre-COVID-19 period, but not all athletes used pre-COVID-19 conventional sports facilities.

The most disturbing consequences of the COVID-19 pandemic, according to Estonian elite athletes, are presented in Figure 3. Looking back at COVID-19 isolation, the athletes deemed the change in training conditions and the cancellation of competitions to be the most disturbing. In June 2021, the biggest concerns were also the cancellation of competitions, as well as the lack of training with companions. In June 2021, compared to 2020, athletes were more deprived of training with their teammates, and the postponement of the Olympic Games was also considered a significant problem.

In total, 80% of the subjects continued to exercise normally, just as they had before COVID-19. However, 65.7% of the respondents found that training was still difficult, and 14.3% considered insufficient cooperation with a coach to be a problem. At the same time, the athletes were satisfied with the availability of health services, and only 18.6% of the respondents complained about their availability to some extent. Emotional support was primarily received from a coach (51.4%), family (28.6), and a close friend or partner (25.7%). 4.3% of the respondents used the help of a sports psychologist as a part of their standard routine. 4.3% of respondents feared becoming infected with the COVID-19 virus and 34.3% were concerned about the possible spread of the coronavirus to their loved ones. The COVID-19 pandemic made 25.7% of the respondents consider ending their careers, while 20% became more invested in sports. The COVID-19 pandemic did not affect the income of 64.3% of the athletes, whereas 31.4% of the athletes admitted that they had problems with sponsors (loss of money, difficulty finding new sponsors, losing a sponsor).

## 4. Discussion

In June 2020 (a couple of weeks after the end of the national COVID-19 lockdown period), the main symptoms of female elite athletes were fatigue and insomnia [13], whereas in June 2021 (more than year after the COVID-19 lockdown period), at the time when the training situation was normalized, the majority of women had symptoms of depression and fatigue. The results of the present study show that despite more openness in the country and the recovery of training conditions, the mental health indicators of elite athletes deteriorated compared to June 2020. The frequency of symptoms indicating mental health problems (depression, anxiety, and fatigue) of female athletes is particularly worrying.

A pandemic is a long and enduring transition period in an athlete’s career, which is also an important stage in the athlete’s development, where stressors must be responded to positively in order to facilitate continued success [19]. Resilience in the face of adversity is a key factor in maintaining mental health [20] in the transition period. However, the latest data show that athletes do not differ in psychological resilience from non-athletes, so they are just as vulnerable as or even more vulnerable than ordinary citizens in a pandemic [4]. It should not be ignored that if the situation becomes better and more stable, then everything will not necessarily be fine for athletes. The athlete must make up for what was missing before, which can make things extremely tense.

The most commonly observed mental health consequences associated with non-normative transitions in sports (e.g., injuries) are depression and anxiety [21]. The COVID-19 pandemic and the resulting isolation (including changes in training conditions, cancellation of competitions, insufficient support from the support system, etc.), as one of the non-normative transitions significantly affected athletes’ depression and anxiety levels. Over the last year, the mental health indicators for Estonian athletes have worsened. Concordantly with other studies, we showed that male and female athletes adapt differently to a pandemic situation [22], and that women are more vulnerable [23,24,25]. This may be explained by the finding by Bowes et al. [11], who found that female athletes considered that they had less access to equipment as women during the COVID-19 pandemic, and that men’s sports were a priority. Women were certainly affected by the reduction in gaming fees and/or sponsorships. The results of our study show that not only the mental health indicators of female athletes were worse than those of men immediately after the lockdown period, but they also became more negative over the year. At the same time, the mental health indicators of male athletes have also deteriorated.

Our study clearly shows that despite the normalization of competition schedules by the summer of 2021 (including the Olympic Games) and the recovery of training conditions (similar to pre-COVID-19), female athletes’ mental health parameters were worse than in June 2020. It is a telling fact that although athletes themselves do not see shortcomings in the availability of a sports psychologist, they do not see a psychologist as their main supporter, and it can be assumed that athletes were not sufficiently trained or supported in this area during the pandemic. It is important to understand that as the symptoms of a mental disorder (such as depression and anxiety) increase, so does the risk of injury [26].

As the restrictions in the COVID-19 pandemic situation varied from country to country, the situations of athletes in each country were not the same nor comparable [21]. However, it can be said that one of the biggest problems for elite athletes in 2020 was the cancellation and postponement of competitions [13,22,27]. The study respondents also noted the cancellation of competitions and the change in training conditions as disturbing factors. During lockdown (2020), out of 12,526 athletes from six continents, only <40% were able to maintain sport-specific training, and most athletes (83%) trained for “general fitness and health maintenance” [27]. In June 2021, most study respondents trained as usual, but they use somewhat less fitness studios and special sports facilities, and undertake more training at home and in the home gardens. This clearly shows that athletes have been able to adapt their homes to their training needs and will probably not need the services of certain training facilities to the same extent in the future. However, the lower use of the swimming pool compared to the COVID-19 period is thought-provoking. This is probably due to the fact that swimming is often used to develop endurance in various sports. Now, alternatives to this have probably been found and are most likely already being used, and so swimming pools are no longer used on a daily basis.

It is worth noting that, although about one-quarter of study respondents considered interrupting their athletic careers because of COVID-19 isolation, approximately the same number of respondents decided to invest even more in sport. This is probably related to how our athletes were able to (re)set their goals in the COVID-19 situation and beyond. Costa et al. [28] highlighted three overarching themes among Italian athletes and coaches in March 2020 in response to goal adjustment,“Moving on toward new goals”, “Letting go of goals”, and “Trying to hold on”. The process goals are important in focusing on key goals and maintaining motivation [29], it must therefore be taken into account by both athletes and their support staff, especially in difficult times such as the COVID-19 pandemic or the current war in Europe.

Closures hindered athletes’ access to their multidisciplinary teams (e.g., coaches, sport science, medical and allied professionals) [30]. At the beginning of summer 2020, 64.7% of Estonia’s elite athletes did not have access to the necessary health care services [14]. Meanwhile, a year after the lockdown period, study respondents considered the availability of health services to be good. However, there was a lack of close cooperation with their coaches, which may have occurred because athletes did not have the opportunity to train with the coach on a daily basis compared to before COVID-19.

As the subjects are relatively young and it can be assumed that their top achievements in sports are still ahead, it is extremely important to provide them with the necessary support system (psychological assistance, adequate cooperation with coaches, financial security). Attention must be paid to the involvement of sports psychologists already in the training process of young athletes so that athletes develop an understanding and the coping skills required during transition periods, which can certainly be stressful. In addition, Di Fronso et al. [31] suggests that athletes use mindful activities that help develop a variety of interoceptive awareness skills and help assess internal bodily signals that are considered critical components of emotional regulation. In the ongoing COVID-19 pandemic situation, it is important to monitor changes over time and react quickly to them. However, nearly a year has passed since the survey was conducted, so the results do not reflect the current situation. As the subjects were certainly at least partially different in 2020 and 2021, it is difficult to assess changes in mental health indicators over time. The fact that the subjects were representatives of different sports and also athletes of different ages can also be considered a limitation of the research. Due to the small number of subjects, we cannot point out differences by sport, which would provide important information for our coaches and associations. However, it is certainly important that we continue to monitor the mental health of our athletes on an ongoing basis and provide them with reliable support when needed.

## 5. Conclusions

More than a year after the COVID-19 isolation lockdown period, the mental health indicators of Estonian elite athletes (especially symptoms of depression and fatigue in females) were worrisome. Most of subjects had high levels of distress, although their training conditions had returned to normal (i.e., as before COVID-19). It takes time to adapt after isolation, and elite athletes may not have enough time to get back to the top of the next Olympic cycle, for example. Training during a period of restrictions and uncertainty about the future is clearly an important factor influencing the mental health of elite athletes.

## Figures and Tables

**Figure 1 sports-10-00076-f001:**
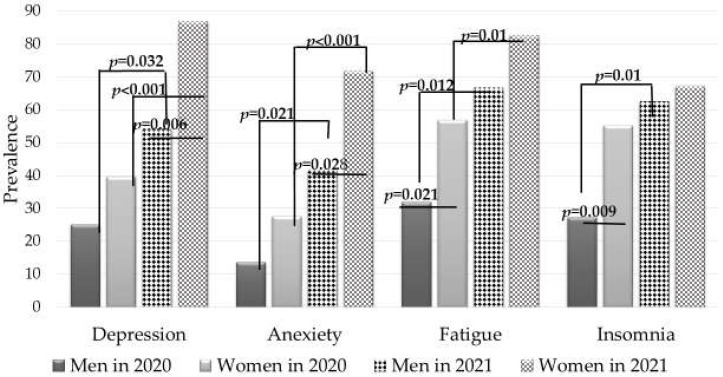
Prevalence of mental health symptoms in elite Estonian athletes a year after the end of the COVID-19 lockdown period (June 2021) compared to June 2020 (Chi-squared test).

**Figure 2 sports-10-00076-f002:**
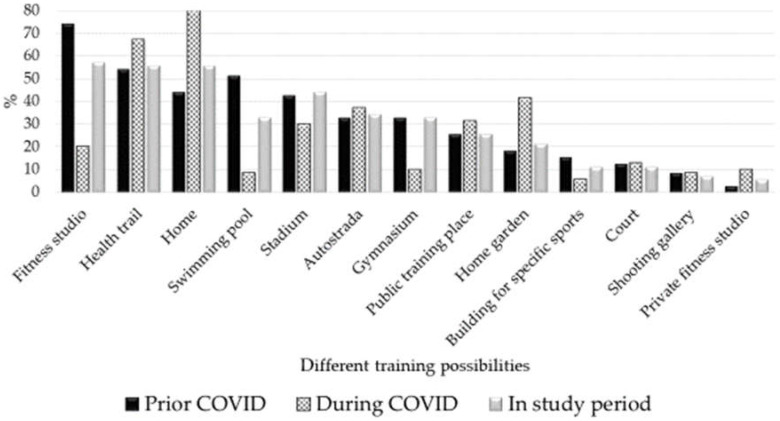
Training sites as assessed by elite athletes prior COVID-19, during COVID-19 lockdown, and a year after the COVID-19 lockdown (June 2021).

**Figure 3 sports-10-00076-f003:**
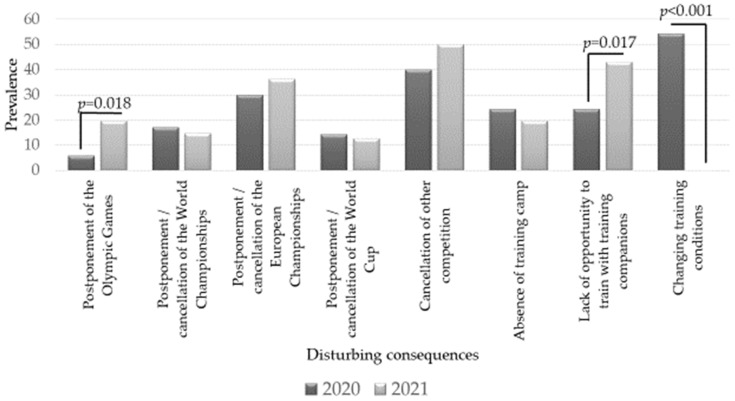
The most disturbing consequences of COVID-19 pandemic according to Estonian elite athletes (Chi-squared test).

**Table 1 sports-10-00076-t001:** Age and gender distribution in different EST-Q2 subgroups.

Year		2020	2021		2020	2021		2020	2021		2020	2021
		*n*		*p*	Female (*n*; %)		*p*	Male (*n*; %)		*p*	Age; Mean (SD)	
Female		58	46								24.7 (8.6)	21.4 (7.1)
Male		44	24								24.8 (8.7)	24.2 (7.4)
EST-Q2	Low distress	33 (32.3)	4 (5.7)	0.001	12 (20.7)	0	0.003	21 (47.8)	4 (16.7)	0.023	24.6 (8.6)	21.5 (3.9)
	Moderate distress	32 (31.4)	10 (14.3)	0.017	15 (25.8)	6 (13%)	NS	17 (38.6)	4 (16.7)	NS	24.8 (8.8)	22.5 (6.1)
	High distress	37 (36.3)	56 (80)	0.001	31 (53.5)	40 (87)	<0.001	6 (13.6)	16 (66.7)	<0.001	24.7 (8.6)	22.4 (7.7)

NS–not significant, *p* indicates significant differences in different level distress groups between 2020 and 2021.

## Data Availability

Not applicable.

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
