# Peer review of "Changes in the Mental Health Indicators and Training Opportunities for Estonian Elite Athletes Compared to the COVID-19 Isolation Period"

_sports, 2022, doi:10.3390/sports10050076_

Round 1

Reviewer 1 Report

The aim of the study was to evaluate the mental health indicators and training opportunities of elite Estonian athletes after the complete COVID-19 isolation period compared to the time during the isolation period.

ABSTRACT: should be improved with more detailed methods and results…

METHODS:

Describe more in detail sample of the study. How you choose all the participants. Inclusion and exclusion criteria…

RESULTS & DISCUSSION

Limitation of the study is missing. Please add at the end of the article the possible limitation of the study. It's not well clarified…

Author Response

Dear reviewer,

thank you for your good suggestions and feedback! We did our best, made a lot of additions and changes, and we hope that the manuscript has improved significantly.

Best wishes

Reviewer 2 Report

Comments and Suggestions for Authors:

Changes in the Mental Health Indicators and Training Oppor-2 tunities of Estonian Elite Athletes Compared to the COVID-19 3 Isolation Period

Abstract. The athletes analysed should be described. Do not repeat words from the title in the key Word.

Introduction. This aspect seems to me to be the weakest aspect of the manuscript. The introduction should be expanded. All study variables should be explained and justified. Which studies use these variables?

Materials and Methods.

Line 59: Signing of informed consent. It should explain the inclusion and exclusion criteria. Athletes under the age of 18? Injured athletes? Proposal for improvement? Ask coaches?

Results. The result and tables are correct. Figures that could be improved.

Line 129: Figure 1: Improve lines indicating significance (p)

Line 149: Figure 3: It is not well understood. It would be better to transfer these results to a table.

Discussion. Line 325, 331, 369- Supplement these statements with the data from the studies cited.

References. Check the format of the references: 2, 4, 16, 24, 32

Making the indicated modifications, the study is novel and interesting.

Author Response

Dear reviewer, thank you for your good suggestions and feedback! We did our best, made a lot of additions and changes, and we hope that the manuscript has improved significantly. 

Best wishes!

Reviewer 3 Report

In this study, the authors analyzed results of the questionnaires responded by the Estonian elite athletes in June 2021 and compared whether they differed from another group of Estonian elite athletes who filled-in same questionnaires in June 2020, immediately after the full lockdown period caused by COVID-19 disease ended. The results are concerning – in comparison with the 2020 survey, one year after the total lockdown 2x more Estonian elite athletes showed symptoms of deteriorated mental health. It is a pity that the authors focused just on differences between sexes, while they could have put more effort to try to identify subgroups that could be under greater stress. The research would be more interesting if authors compared stress between athletes who got COVID disease (77%) with those who did not get it, as well as between athletes who play individual sports (77.1%) with those who participate in group sports and can moderate stress thanks to the support of their peers.

Major objections:

In Table 1, the authors presented the exact number and prevalence of participants in the low, moderate and high stress subcategories defined according to the EST-Q2 questionnaire. In lines 102-105, ORs are given for the possibility of belonging to these three stress subgroups in 2021, when compared to 2020 – the ORs for moderate and low distress subgroups differ from the results I got using online odd ratio calculator. The authors should recalculate the ORs.

Minor objections:

The authors should use one term when indicating which of the two surveys they refer to in the manuscript – for instance, in line 34 they report on the „In spring 2020“ survey, while in line 60 the same survey is referred to „the study published in 2021“. Since in the manuscript authors are also present results of the 2021 survey, please use the same name continuously for each study.  

The authors often refer to the June 2021 survey as the one „after the end of the COVID-19 isolation period“, while the total lockdown in Estonia ended of May 17th 2020, after which the May-June 2020 survey was conducted. So unless there were some other isolation periods after the 2020 total lockdown, and if there were you should indicate when they were and how long did they last, please use the term „the 2021 survey“ or „one year after the total lockdown survey“.

Other comments

Title

line 3 – please replace „opportunities of“ with „opportunities for

Abstract

Please indicate how many respondents were in your study, and what share of the total number of Estonian elite athletes they represent. Also state that both surveys you compared were cross-sectional. The authors wrote that their aim was to compare the 2021 survey results with the 2020 survey results, but in the Results section of the Abstract they did not statistically compare old and new results.

line 12 – please replace „opportunities of“ with „opportunities for

lines 12-13 It is not clear which periods the authors compare when they say „after the complete COVID-19 isolation period, compared to the time during the isolation period“? This study brings the results of the 2021 survey (conducted in June 2021) and compares it with the 2020 survey (conducted from May to June 2020, but covers the COVID-19 lockdown period).  I checked, the state of emergency (total lockdown) in Estonia lasted from March 13 to May 17, 2020. Thus, the June 2021 survey cannot be defined as "after the complete COVID-19 isolation period", because it was actually conducted one year after complete isolation, and not immediately after the total lockdown in 2020. Or if the “total isolation COVID-19 period” you refer to lasted longer, state which exact period you mean.

line 14 – please replace „X2“ with „Chi2

line 15 – again, the authors refer to „Despite the end of national COVID-19 period“ – it would be more clear if they would use terms like „More than a year after the COVID-19 lockdown period (or in the 2021 survey) …. the mental health problems of Estonian elite athletes are of even greater concern than they were at the time of the COVID-19 lockdown.“

Introduction

This section is very short, the authors cite only four papers.

line 46 – please add „their long-term mental health“

line 48 „athletes a year after the complete COVID-19 …“

Materials and Methods

How many minors were in this study?

line 57 – please change the year in „beginning of June 2021“

line 60 – please change „the study published in 2021“ in „the study conducted in 2020

lines 67- 68 – it is not clear what does it mean to „identify persons with mental health symptoms“. Maybe you can say „symptoms related to mental health“ or „symptoms of mental health problems“?

lines 68-69 – „a survey conducted in June 2020, after the isolation period, but relating to the COVID-19 isolation period.“

line 78 – please replace „X2“ with „Chi2

Results

line 83 – please change to „102 in 2020 and 70 athletes in 2021.“

line 85 – „… but we have presented some of these results here as well to determine if there has been a change between …“

line 86 – please replace „the participants group constituted“ to „respondents represented ~12% of all invited participants (Estonian elite athletes)

line 87 – change „y“ to „years“

Lines 89-90 - if you decided to show the exact prevalence of athletes, swimmers and skiers, then you should categorize the remaining 61% of the respondents as well

line 96 – please change the subtitle according to the suggestions I gave for lines 67-68

lines 97-108 – This paragraph describes the results of Table 1, in which comparisons between sexes and surveys are shown. Therefore, it is not enough to write that „In 2021, 80% of the subjects had high distress“ unless you compare it in the same sentence to the 2020 survey results (you can, for instance, add „, which is more than twice as much as in the 2020 survey“.

Furthermore, you should describe prevalence of athletes in the EST-Q2 subgroup, and not just write „numbers of athletes“.

line 106 – „Among athletes, there were significantly more women who …“

Table 1 – please explain in the legend what * and ** (used in the second and the fifth column) mean

line 121 – „symptoms related to mental health“

Subparagraph 3.3. Training Conditions, lines 134-141

While in 2021 the training opportunities were normalized compared to the COVID-19 lockdown period, I don`t see that all the athletes returned to their old habits. The situation before the lockdown differed from the one in 2021 for many training locations - approx. 73% vs. 57% reported training in fitness studio, approx. 44% vs. 55% reported training at home, approximately 51% vs. 32% reported training at the swimming pool. Why? Can any conclusion be drawn?

The authors did not comment on these results in the Discussion section.

line 158 – did the athletes seek counseling with a psychologist or was it part of a standard routine?

lines 159 -160 - were the athletes who worried about their health the ones who had COVID-19 disease, so they were afraid of the long-term consequences of COVID, or were they afraid of getting COVID?

Discussion

The authors repeat themselves (lines 167-169, then again in lines 202-204).

lines 175-176 – what did you want to say with this fragment of the sentence?

„A pandemic is a long and enduring transition period in an athlete's career, which is also an important stage in the development of a sport …“

line 196 please replace „although“ with „not only

line 197 please add „ but they also became …“

Author Response

(The authors gave the same response as above.)

Round 2

Reviewer 3 Report

The manuscript has been substantially improved and the errors have been corrected. I have only two minor remarks:

Abstract, line 20 – please replace „in comparing“ with „compared to“

Introduction, line 34 – please delete „probably“

Author Response

Thank you, the corrections are done.

Best wishes!